# Coccolithophore assemblage composition during the Greenland Interstadial–Stadial 20 transition and their response to the Youngest Toba Tuff (YTT) supereruption ∼74,000 years ago in the northeastern Arabian Sea

**Jose Dominick Guballa**[1]*, **Jörg Bollmann**[1], **Katherine Schmidt**[1], **Andreas Lückge**[2]

**1** Department of Earth Sciences, University of Toronto, Toronto, Ontario, Canada, **2** Bundesanstalt für Geowissenschaften und Rohstoffe (BGR), Hannover, Germany

\* doms.guballa@mail.utoronto.ca

## Abstract

Here we present a new 850-year long coccolithophore record from core SO130-289KL in the northeastern Arabian Sea that spans the Greenland Interstadial 20–Greenland Stadial 20 transition including the timing of the ∼74 ka Youngest Toba Tuff (YTT) supereruption. During the warm interstadial, the coccolithophore assemblage is characterized by *Gephyrocapsa oceanica* (41%) and *Florisphaera profunda* (30%) while the succeeding cold stadial is distinguished by the abundance of small placolith species (*Emiliania huxleyi* + *G. ericsonii*) (52%). However, the oldest samples of the study interval seem to show a stadial-like coccolithophore assemblage. Spectral analysis revealed an interdecadal cycle imprinted in the coccolithophore record during the interstadial that was also independently reported in other terrestrial and marine proxies. Immediately after the YTT eruption, small placoliths increased by 42% from 5% right below the YTT layer to 47% ∼1 mm above the YTT layer, while *G. oceanica* and *Helicosphaera carteri* increased and *F. profunda* decreased within the ∼1.15 cm layer representing about 8–19 years. Subsequently, the coccolithophore assemblage returned to a composition similar to the warm interstadial period before abruptly changing to a stadial composition characterized by the abundance of small placoliths about 100–130 years after the YTT eruption. Therefore, the YTT eruption had a significant impact on the overall coccolithophore assemblage but appears to have not caused the climate transition from interstadial to stadial conditions, supporting proxy and modeling data. However, the overall mechanism driving the observed changes and cyclicities remains unknown but might be related to rapid atmospheric teleconnections of North Atlantic climate variability to the low latitudes.

**Data Availability Statement:** All relevant data are within the manuscript and its Supporting Information files.

**Funding:** This work was funded by NSERC through J. B.'s NSERC Discovery Grant [http://www.nserc-crsng.gc.ca]. NSERC had no role in the study design, data collection and analysis, decision to publish, or preparation of the manuscript.

**Competing interests:** The authors have declared that no competing interests exist.

## Introduction

Anthropogenic greenhouse gas emissions are currently changing climate at an unprecedented rate, impacting ecosystems on a global scale [1]. A natural process capable of such rapid changes are bolide impacts [2] which have driven evolution [3, 4] and climate change [5, 6] in Earth's history. However, the effects of rapid environmental events such as volcanic supereruptions on climate [7–10], evolution, and the marine ecosystem, especially on marine phytoplankton, are not well constrained.

For example, the most recent supereruption of the Toba volcano, the Youngest Toba Tuff (YTT) eruption ∼74 ka [11–13], is the largest supereruption in the Quaternary [14] (Fig 1) but the impact of the eruption on both human evolution [15–17] and global climate [7–10] remain disputed. The YTT eruption was thought to have initiated a global volcanic winter, resulting in the transition from a warm interstadial (Greenland Interstadial 20 or GI-20; [18]) to a cold, stadial climate (Greenland Stadial 20 or GS-20) [9, 19]https://www.zotero.org/google-docs/?O86JiR. These climate transitions of the millennial-scale Dansgaard–Oeschger (D–O) oscillations [20] are characterized by abrupt (i.e., decadal-scale) warming and relatively more gradual (i.e., multidecadal to centennial) cooling transitions [18, 21], with the latter recently garnering new attention [22] in the context of understanding climate "tipping points" [23].

The estimated magnitude of global cooling due to $SO_4$ emissions by the YTT eruption in models ranges considerably, from ∼8˚C [10] to ∼2–4˚C [7, 25] and, more recently, a maximum cooling of up to 1.5˚C to a possible warming scenario of up to 2˚C [8]. However, in contrast to a homogeneous impact, recent modeling studies argue for a globally heterogeneous climatic effect of the YTT eruption [7, 25]. For instance, positive excursions in the stable oxygen isotope composition ($\delta^{18}O$) of speleothems [26, 27] and sea surface temperature (SST) cooling of 1˚C based on alkenones ($U^K_{37}$) in the South China Sea [28] suggest a drying and cooling effect, respectively, of the YTT eruption. In contrast, lake records from Africa [29, 30] and SST from alkenones in the Arabian Sea [31] indicate minimal to no climatic effects.

More recently, a strengthened Indian winter monsoon after the YTT eruption was proposed to have driven a significant increase in marine primary productivity (PP) in the northeastern Arabian Sea as inferred from the decrease in the relative abundance of the lower photic zone coccolithophore species *Florisphaera profunda* [11]. However, the study only focused on one coccolithophore species for the PP reconstruction, and the impact of the YTT eruption on the overall coccolithophore assemblage has not yet been analyzed even though coccolithophores have been reported to be sensitive to changes in surface water properties [32–35].

Here we present a new dataset on the coccolithophore assemblage composition that spans the GI-20 to GS-20 transition including the timing of the YTT eruption from highly temporally resolved laminated sediments from the northeastern Arabian Sea. This dataset provides new insights on how geologically instantaneous events such as volcanic eruptions as well as climate tipping points such as D–O transitions have impacted coccolithophores in the past, a prerequisite to assess the response of this important marine phytoplankton group to the current climate crisis.

## Materials and methods

Core SO130-289KL (20.2 m long; 23˚ 07.3' N, 66˚ 29.8' E) was recovered at 571 m water depth northwest of the Indus River submarine canyon [36] within the center of a stable oxygen minimum zone at the continental slope of the northeastern Arabian Sea [37] (Fig 1). The lack of oxygen at the core location allowed for the preservation of laminated interstadial sediments of high temporal resolution and the interstadial–stadial transitions are well defined, including the GI-20 to GS-20 transition [38, 39]. Furthermore, two ash layers established to be correlated

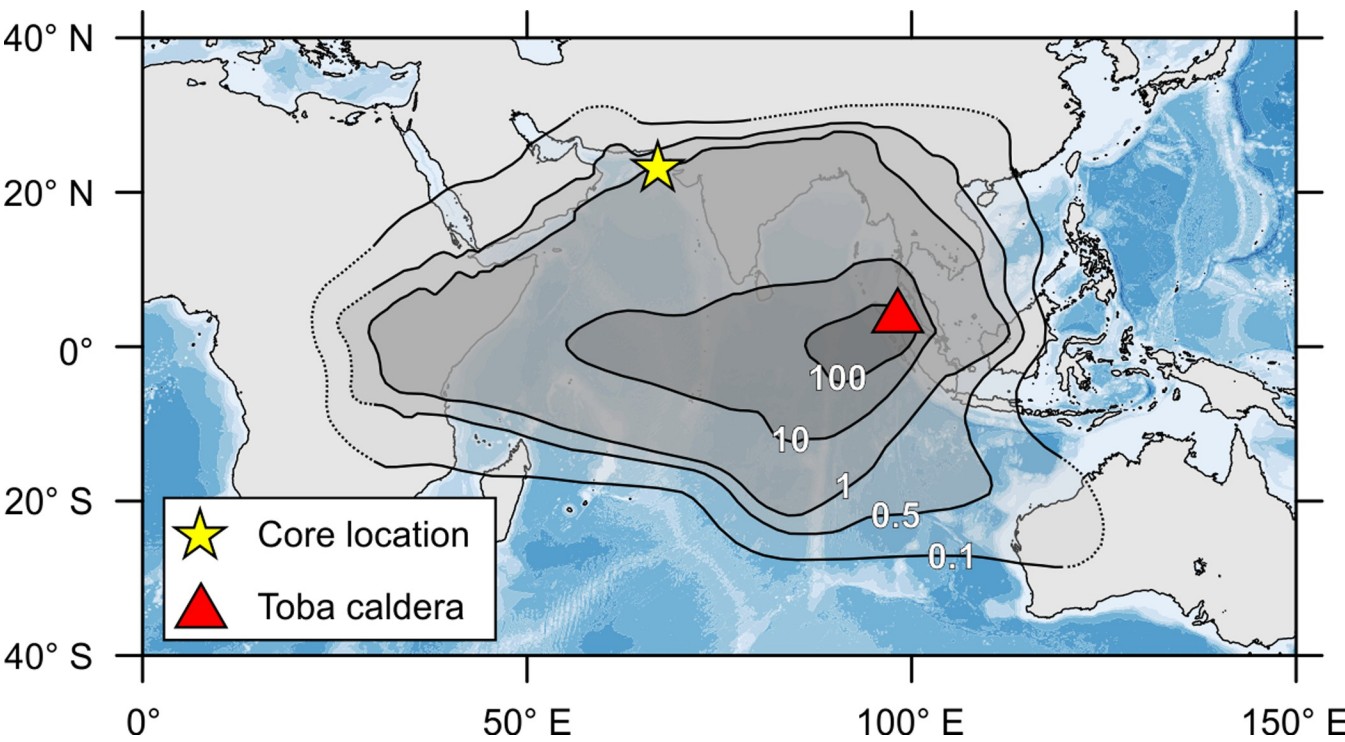

**Fig 1. Extent of the Youngest Toba Tuff (YTT) eruption with respect to the location of core SO130-289KL.** The black contour lines represent the projected ashfall thickness in centimeters based on modeling [24]. Dashed lines are estimated extent of the ashfall based on the form of the contours. The yellow star approximates the location of core SO-130-289KL. The base map was made using data freely available from the General Bathymetric Chart of the Oceans (GEBCO) Grid website (https://www.gebco.net/data_and_products/gridded_bathymetry_data/).

to the YTT eruption enclose ∼1.15 cm of laminated sediments as part of a miniature slump fold [39], preserving environmental conditions immediately after the YTT eruption.

A ∼1 cm-wide U-channel sampler was used to obtain a subsection between 1870 to 1820 cm core depth (hereafter referred to as cm), which covers the interval before and after the YTT eruption and the transition from Greenland Interstadial 20 to Greenland Stadial 20 [38]. We focused on the interval between 1854.6–1820.4 cm, corresponding to a time period between 74.353 to 73.505 ka with a sampling resolution between 2–21 years and an age estimate of 73.9 ± 0.1 ka (2σ) for the YTT eruption [11]. The 1.15 cm-thick fold between the two YTT layers represents a duration of approximately 8–19 years based on various assumptions of sediment repetition [11].

No permits were required for the described study, which complied with all relevant regulations.

## Coccolithophore analysis

A total of 107 samples were analyzed for coccolithophore assemblage composition and the sampling strategy and sample preparation are described in Guballa et al. [11]. Samples were analyzed using a Zeiss Axio Imager Z1 transmitted light microscope at 1250x magnification using a PlanApo oil immersion objective (numerical aperture = 1.4) and circularly polarized light [40]. We used standard taxonomic descriptions for coccolithophore species identification [41, 42]. At least 300 coccoliths (i.e., isolated calcareous plates of coccolithophores) were counted per sample to calculate the relative abundance of each species and the corresponding 95% confidence intervals were calculated using the Wald interval [43]. We combined the

coccolithophore species *Emiliania huxleyi* and *Gephyrocapsa ericsonii* as "small placoliths", as the minute sizes of these species made identification of some specimens on the light microscope difficult. Furthermore, the different species of *Umbilicosphaera* (i.e., *U. sibogae* and *U. foliosa*) were lumped together as *U. sibogae*. Relative abundances of the species *F. profunda* and *Helicosphaera carteri* are taken from Guballa et al. [11]. All data generated from this study are provided in S1 Table.

## Spectral analyses

Spectral analysis was done on the unevenly spaced data set using the "redfit" function [44] in the dplR package version 1.7.2 [45] in R v.4.3.3 [46]. This function avoids linear interpolation that artificially decreases the spectral power of higher frequency components [47]. The analysis was applied to samples prior to the YTT eruption (i.e., 1854.6–1843.3 cm, 74.353–73.886 ka) using coccolithophore species that were consistently present throughout the studied interval. To calculate the spectrum, the default settings were applied except for the number of segments where we only used one due to the relatively few points in the interval (n = 60). Significance testing was performed using the same function by running 2000 Monte Carlo simulations to compute for the 90%, 95%, and 99% red noise false alarm levels.

## Results

Core SO130-289KL from the northeastern Arabian Sea is composed of dark-colored, indistinctly to distinctly laminated, carbonate-poor, and organic-rich interstadial sediments from 1854.6–1835 cm core depth and light-colored, homogeneous, bioturbated, carbonate-rich and organic-poor stadial layers from 1835–1820 cm [38, 39, 48] (Fig 2A and 2B). The YTT eruption is recorded in the core as two discrete, ∼1 mm thick ash layers found at 1843.2 and 1842.05 cm enclosing ∼1.15 cm of laminated sediments interpreted as a miniature slump fold [39]. Within these sediments, twenty-one (21) coccolithophore species were identified in 107 samples between 1854.6–1820.4 cm (∼850-year duration) (S1 Table) and coccolith specimens are generally well-preserved (S1 Fig). Around 90% of the coccolithophore assemblage is comprised of six species: *G. oceanica*, small placoliths belonging to *E. huxleyi* and *G. ericsonii*, *U. sibogae*, *F. profunda*, and *H. carteri* (Fig 2).

## Coccolithophore assemblage composition during Greenland Interstadial 20 (GI-20)

The coccolithophore assemblage from 1854.6–1843.3 cm (∼470-year duration) falls within Greenland Interstadial 20 (GI-20). During this time interval, *G. oceanica* is the most common species with an average relative abundance of ∼41 ± 5% (Fig 2C). It increases from an average of ∼24 ± 5% to an average of ∼45 ± 5% around 400 years prior to the eruption and shows high variability until before the YTT eruption. (Fig 2C). Small placoliths (*E. huxleyi* + *G. ericsonii*) are highest (∼45 ± 5%) between 470–400 years before the eruption and subsequently remain relatively low (15 ± 4%) until before the eruption (Fig 2D). The species *U. sibogae* varies between ∼0.3–19% and peaks around 440–430 years before the eruption (Fig 2E). After a decline between 300–200 years prior to the YTT eruption, *U. sibogae* subsequently returns to previous values except for sharp declines just before the eruption (Fig 2E). Although *H. carteri* is a minor component of the coccolithophore assemblage (<3%), it seems to have a general pattern of occurrence: it is found between 440–300 years before the eruption, undetected from 300–200 years before the eruption, and then reappears between 200–25 years before the eruption (Fig 2F). *Florisphaera profunda* contributes an average of 30 ± 5% to the assemblage, reaching up to 55–60% centered at 250 and 75 years prior to the eruption (Fig 2G).

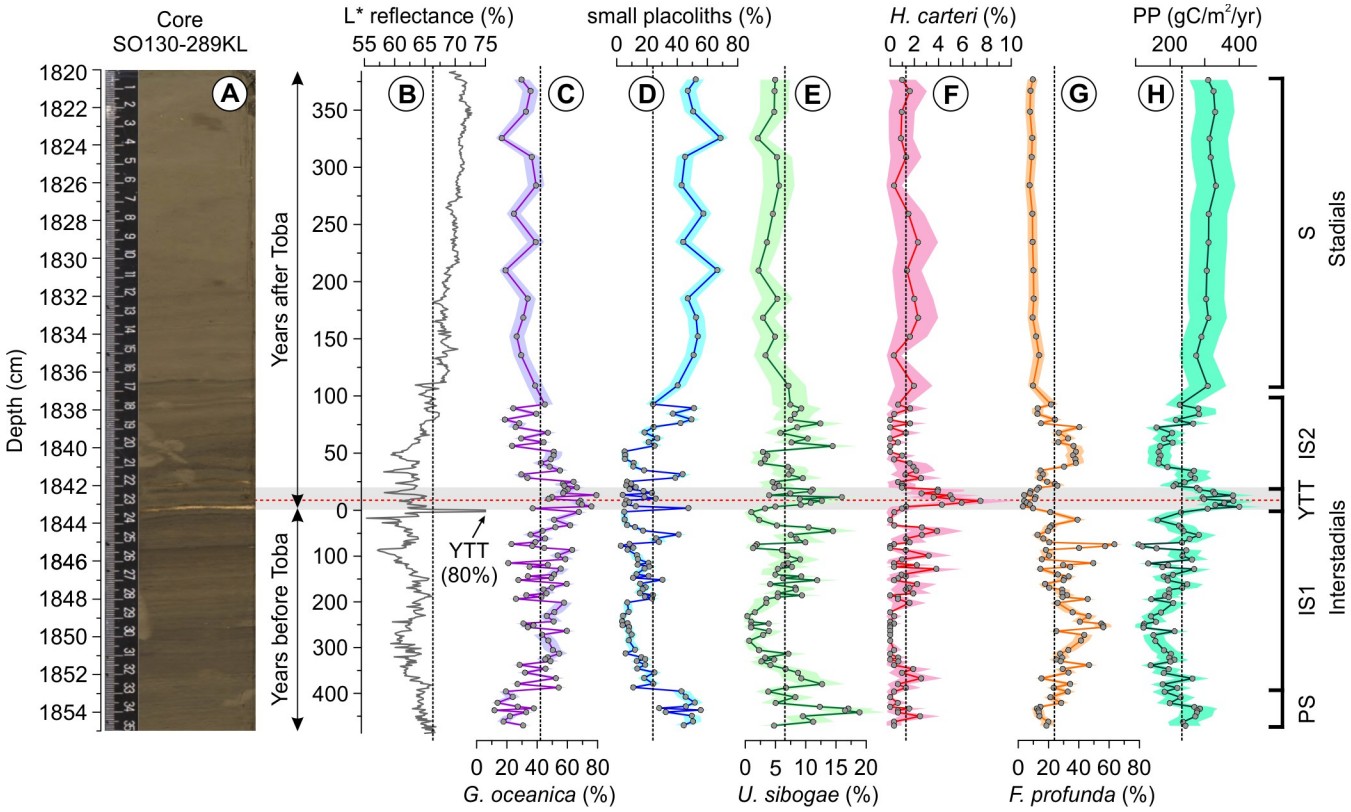

**Fig 2. Relative abundance (%) of the main coccolithophore species between 1854.6–1820.4 cm (74.353–73.505 ka) of core SO130-289KL.** Data are plotted in relative ages before and after the YTT eruption represented by the lower YTT layer (73.9 ka). (A) Sediment core photo from Deplazes et al. [49] showing clearly visible YTT layers. (B) Sediment L* reflectance [50] showing the lower YTT layer peak ($\sim$80% L* reflectance) at 1843.2 cm. (C) *G. oceanica*. (D) small placoliths composed of *E. huxleyi* and *G. ericsonii*. (E) *U. sibogae*. The data for graphs F–H are taken from Guballa et al. [11]: (F) *H. carteri*. (G) *F. profunda*. (H) Annual marine primary productivity (PP) (gC/m²/yr) based on a single coccolithophore species transfer function [51]. The black dotted lines in B–H show the mean value over the entire study interval. Color bands in C–G demarcate the 95% confidence interval of the data. The gray band shows the miniature slump fold enclosed by two ash layers from the YTT eruption, with the red dotted line delineating the position of the axial trace [11]. Labels at the far right show informal intervals based on changes in the coccolithophore assemblage composition. Abbreviations: PS = "pseudo-stadial", IS1 = interstadial before the YTT eruption, YTT = Youngest Toba Tuff, IS2 = interstadial after the YTT eruption, S = stadial.

In general, visual examination of the record during the interstadial shows a shift in the assemblage composition at 1852.8 cm, about 400 years before the YTT eruption (Fig 2). Between 470–400 years before the YTT eruption (1854.6–1852.8 cm), the interstadial assemblage is characterized by small placoliths as the most abundant group. In contrast, the remaining section of the interstadial before the YTT eruption (1852.6–1843.2 cm) is characterized by *G. oceanica* and *F. profunda* being more abundant than small placoliths.

## Coccolithophore assemblage composition immediately after the YTT

In comparison to the interstadial assemblage composition from 1854.6–1843.3 cm, *G. oceanica* increases from a mean value of 41 ± 5% to 60 ± 5% with maximum values of up to 80% within the miniature slump fold (1843.1–1842.1 cm) except for one sample 1 mm above the YTT layer (Fig 2C). In this sample (1843.1 cm) about $\sim$2 years after the YTT eruption, the relative abundance of *G. oceanica* is significantly lower (37 ± 5%) compared to the mean (60 ± 5%) in this interval (Fig 2C). In contrast, small placoliths sharply increase from 5 ± 2% at 1 mm below the YTT layer to 47 ± 6% at 1 mm above the YTT layer, subsequently returning to low values (5–25%) (Fig 2D). *Umbilicosphaera sibogae* reaches a maximum of 16 ± 4% in the miniature

slump fold (Fig 2E), and *H. carteri* increases up to 7 ± 3% which shows an approximately symmetrical relative abundance pattern within the miniature slump fold (Fig 2F). In contrast, *F. profunda* significantly decreases from an average of 30 ± 5% before the YTT eruption to an average of 8 ± 3% right after the YTT eruption (Fig 2G).

## Coccolithophore assemblage composition after the YTT and during Greenland Stadial 20 (GS-20)

The overall coccolithophore assemblage returns to an "interstadial" composition (i.e., 1852.6–1843.2 cm) mainly composed of *G. oceanica* and *F. profunda* (Fig 2) between 20–100 years after the YTT eruption (1840.9–1836.6 cm) before being replaced by small placoliths between 100–130 years after the YTT eruption (1836.6–1835 cm). This major shift also coincides with the transition from laminated dark-colored to homogeneous light-colored sediments, characterizing the succeeding cold Greenland 20 Stadial (Fig 2A and 2B). Samples in the stadial interval ∼ 130–370 years after the YTT eruption (1835–1820.4 cm) are characterized by higher relative abundances of small placoliths (45 ± 5%) than *G. oceanica* (30 ± 5%) (Fig 2C and 2D) and low average relative abundance of *U. sibogae* (4 ± 2%; Fig 2E), *H. carteri* (1 ± 1%; Fig 2F), and *F. profunda* (10 ± 3%; Fig 2G). Variations in the overall coccolithophore assemblage in this interval are clearly driven by changes in *G. oceanica* and the small placoliths (Fig 2C and 2D).

## Spectral analysis

Spectral analysis of our data set was done prior to the YTT eruption (i.e., 1854.6–1843.3 cm, 74.353–73.886 ka) since (1) sediments above the ash layer at 1843.2 cm might be disturbed by slumping [39] and may not necessarily represent the correct age and (2) the coccolithophore assemblage right after the YTT eruption (i.e., in the ∼ 1.15 cm interval) might influence the long-term spectral signal and may skew the resulting spectrum. The analysis reveals a 20–22 and 20–21-year spectral band that exceed the 95% confidence level in *G. oceanica* and small placoliths, respectively (Fig 3A and 3B). In contrast, the power spectrum of *U. sibogae* reveals a significant 28–29-year band (Fig 3C). The power spectrum of *F. profunda* does not show any significant spectral bands at the 90% confidence level although the 18–19-year band almost reaches the 90% confidence level threshold (Fig 3D).

## Discussion

Core SO130-289KL was obtained from a water depth (571 m; [39]) that is significantly above the average lysocline depth (3,300 m; [52] https://www.zotero.org/google-docs/?99sgZM) and we observed that the coccoliths in our samples are mostly well preserved (S1 Fig). Hence, taphonomic effects (i.e., carbonate dissolution) did not cause significant alterations to our data and variations in the coccolith assemblage reflect changes in surface water conditions in the northeastern Arabian Sea. Indeed, this interpretation is supported by combined plankton, sediment trap, and surface sediment analyses in the area which showed that the proportion of the main coccolithophore species in the water column and in the sediments are statistically similar [53].

The coccolithophore assemblage composition in core SO130-289KL compares well with several downcore Holocene and Pleistocene records from the Arabian Sea [54–57] where *G. oceanica*, *F. profunda*, and small placoliths composed of *E. huxleyi* and *G. ericsonii* comprise the major proportion of the coccolithophore assemblage.

However, our record reveals new insights on changes in the coccolithophore assemblage composition immediately after the YTT eruption and in the interstadial–stadial transition (Fig 2). As coccolithophores have been reported to be sensitive to changes in surface water properties [32–35], we assume that our coccolithophore data from the northeastern Arabian Sea

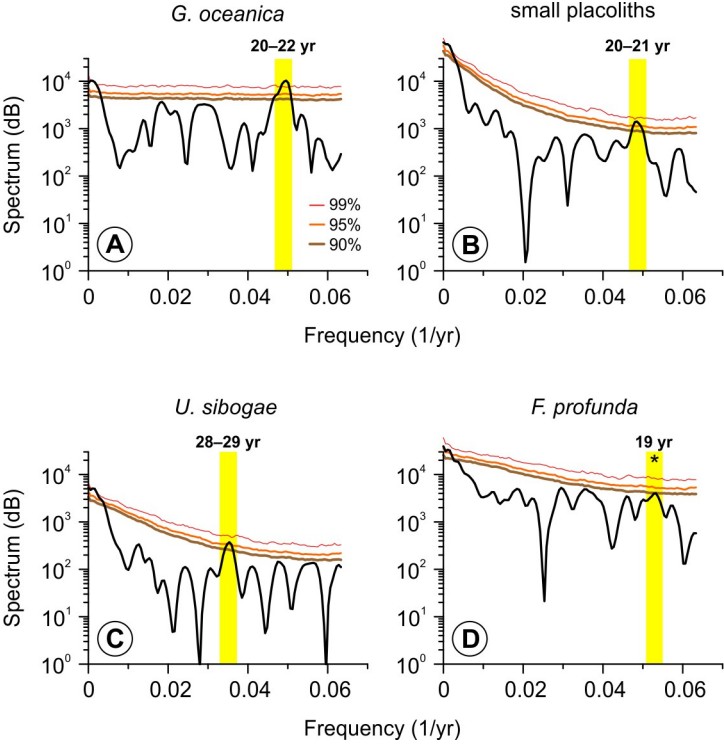

**Fig 3. Spectral analysis of the relative abundance record of the main coccolithophore species.** (A) *G. oceanica*. (B)
small placoliths consisting of *E. huxleyi* and *G. ericsonii*. (C) *U. sibogae*. (D) *F. profunda*. The black line shows the
spectrum for each species. Brown, orange, and red lines represent the 90%, 95%, and 99% confidence limits,
respectively. The yellow rectangles highlight bands exceeding the 90% confidence limit. Yellow rectangles with
asterisks show bands that are very close to the 90% confidence limit.

reflect the impacts of major environmental perturbations related not only to the YTT eruption,
but also to the climatic transition from Greenland Interstadial 20 (GI-20) to Greenland Stadial
20 (GS-20) and potentially to climate oscillations during GI-20.

The present-day climate in the Arabian Sea is primarily driven by differential heating of the
Asian Plateau and the Southern Indian Ocean, resulting in seasonally reversing monsoon
winds [58]. During boreal summer (June–September), faster heating of the Asian Plateau
forms a lower air pressure over the continent relative to the Southern Indian Ocean. This air
pressure gradient leads to the warm and humid cross-equatorial jet of the southwest monsoon
[59] that drives a clockwise basin-wide surface circulation in the Arabian Sea [60, 61] (Fig 4A
and 4B). These surface currents and wind stresses cause upwelling along the coasts of Somalia
and Oman and promotes primary productivity (PP) in the northwestern Arabian Sea [62, 63].
Coastal upwelling also occurs along the coast of Pakistan, on the west side of the northeast–
southwest trending Murray Ridge [64]. In general, the upwelling is restricted to the west of the
ridge, thus there is relatively low summer PP in the northeastern Arabian Sea east of the Mur-
ray Ridge where our core is located [64, 65] (Fig 4A).

During boreal winter (December–February), the air pressure gradients reverse, leading to
dry and cold northeasterly winds of the Indian winter monsoon. These winds induce a less
prominent counterclockwise basinal surface circulation [60, 61] (Fig 4C) and cool sea surface
temperatures by ∼5°C from ∼29°C to ∼24°C [69]https://www.zotero.org/google-docs/?
wzgNCp. The cooling leads to densification and associated convection of surface waters,
resulting in a deepening of the mixed layer [70, 71] (Fig 4D). The mixed layer deepening

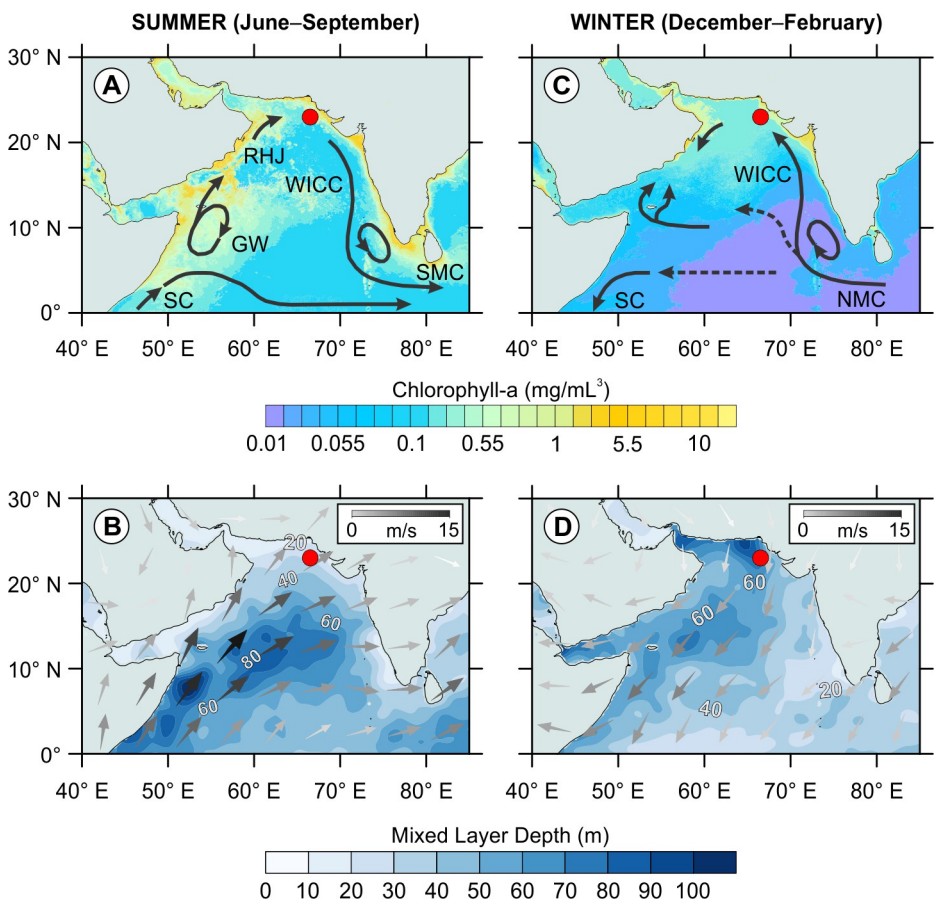

**Fig 4. General seasonal hydrography of the northern Arabian Sea.** The red dots in each panel show the location of core SO130-289KL. (A) Summer (June–September) surface ocean circulation [60, 61] superimposed on average chlorophyll concentrations (mg/mL[3]) between 2005–2017 [66] as a proxy for primary productivity. (B) Summer average surface wind vectors (m/s, arrows) between 2005–2017 [67] overlain on mixed layer depth (m, contours) between 2005–2017 [68]. (C) As in (A), but for winter (December–February). (D) As in (B) but for winter. Abbreviations: GW = Great Whirl, NMC = Northeast Monsoon Current, RHJ = Ras al Hadd Jet, SC = Somalia Current, SMC = Southwest Monsoon Current, WICC = West India Coastal Current. The chlorophyll concentration maps were made using data freely available in the Giovanni online data system (https://giovanni.gsfc.nasa.gov/giovanni/) [66], developed and maintained by the National Aeronautics and Space Administration Goddard Earth Sciences Data and Information Services Center (NASA GES DISC). The chlorophyll data are derived from Moderate Resolution Imaging Spectroradiometer on the Aqua satellite (MODIS-Aqua) provided to Giovanni by the Ocean Biology Distributed Active Archive Center (OB.DAAC). The mixed layer depth maps were generated using data from the World Ocean Atlas (WOA) 2018 data [68] publicly available from the National Centers for Environmental Information (NCEI) of the National Oceanic and Atmospheric Administration (NOAA) website (https://www.ncei.noaa.gov/data/oceans/woa/WOA18/DATA). The wind vector data from the National Centers for Environmental Prediction/ National Center for Atmospheric Research (NCEP/NCAR) Reanalysis Project [67] are freely available from the Physical Sciences Laboratory of the National Oceanic and Atmospheric Administration (PSL-NOAA) website (https://psl.noaa.gov/cgi-bin/data/composites/printpage.pl).

entrains nutrients from the subsurface nutricline towards the surface and ultimately increases basin-wide PP in the northern Arabian Sea, including the location of core SO130-289KL (Fig 4C and 4D).

Sediment trap [72, 73] and plankton studies [74, 75] have revealed that coccolithophores in the Arabian Sea respond to variations in the strength of the monsoons and associated changes in the surface ocean mixed layer depth. For example, the decrease of the lower photic zone species *F. profunda* in core SO130-289KL suggests a deepened mixed layer due to a strengthened

Indian winter monsoon that leads to an increase in PP in the northeastern Arabian Sea [11]. Similarly, our data show that the upper photic zone species *G. oceanica* and *H. carteri* reach their highest relative abundance values (80% and 7%, respectively) right after the YTT eruption (Fig 2) concomitant with the proposed deepening of the mixed layer [11]. Subsequently, the coccolithophore assemblage returns to previous "interstadial" compositions characterized by *G. oceanica* and *F. profunda*, lasting for a few decades before shifting towards an assemblage mainly composed of small placoliths (i.e., *E. huxleyi* and *G. ericsonii*) in the succeeding stadial (Fig 2).

## Coccolithophore assemblage changes and interstadial–stadial transitions

The impact of the YTT eruption on the GI-20 to GS-20 transition has been heavily debated on whether it triggered the climate cooling [9, 19, 76] or only contributed to the cooling [77, 78]. In this context, the fact that our data clearly show a striking change in the coccolithophore assemblage immediately after the YTT eruption (Fig 2) strongly suggests that the eruption significantly impacted the marine environment. However, the return of the coccolithophore assemblage to "interstadial" compositions before transitioning to "stadial" compositions supports the view that the YTT eruption was not the trigger for the transition from GI-20 to GS-20 [10, 77, 78].

During the interstadial (GI-20) to stadial (GS-20) transition, the change in the coccolithophore assemblage composition from *G. oceanica* and *F. profunda* to small placoliths composed of *E. huxleyi* and *G. ericsonii* aligns with many other proxies that recorded this climatic transition (Fig 5). Indeed, the assemblage shift also follows the change in sediment texture from distinctly laminated, dark-colored interstadial sediments to homogeneous, bioturbated, light-colored stadial sediments also reflected by the increasing trend of L* reflectance at the GI-20 to GS-20 transition (Fig 5A and 5C) interpreted as a weakening of the Indian summer monsoon [38, 79]. These changes follow within age uncertainties a positive shift in the oxygen isotopic composition ($\delta^{18}O$) of speleothems in China suggesting a weakened Asian summer monsoon [27, 80] (Fig 5D), associated with a southward migration of the Intertropical Convergence Zone (ITCZ) [81–83] and decreased air temperatures recorded in Greenland ice cores [21, 84] (Fig 5E). On the other hand, increasing median grain sizes of Chinese loess deposits, a proxy for winter monsoon strength, most likely indicates a strengthened winter monsoon system induced by the interstadial–stadial transition [85–87]. Hence, the alignment of our data with many other proxies (Fig 5) shows the potential of using coccolithophore assemblage changes to trace Dansgaard–Oeschger (D–O) climate transitions in middle to low latitudes.

## Stadial-like coccolithophore assemblage during an interstadial?

We also detected significant changes in the coccolithophore assemblage that do not seem to be reflected in other proxy records. For instance, within the interstadial prior to the YTT eruption, the coccolithophore assemblage is characterized for about 70 years (470–400 before the YTT) by *G. oceanica* and small placolith abundances very similar to the succeeding "stadial" assemblage composition (Fig 6). The change in the coccolithophore assemblage aligns well with L* reflectance and primary productivity (Fig 5A–5C) but not with other proxy records (Fig 5D and 5E). We tentatively called this time period a "pseudo-stadial".

It is crucial to note that our record is close to the transition from Marine Isotope Stage (MIS) 5a to MIS 4 ~71 ka [89], characterized by rapid and extensive reorganization of the climate [90] and $CO_2$ drawdown [91]. Although further independent evidence is required, we speculate that a transient strengthening of the winter monsoon near the end of a warm interstadial as the climate system started to shift from one mode (i.e., last interglacial period) to

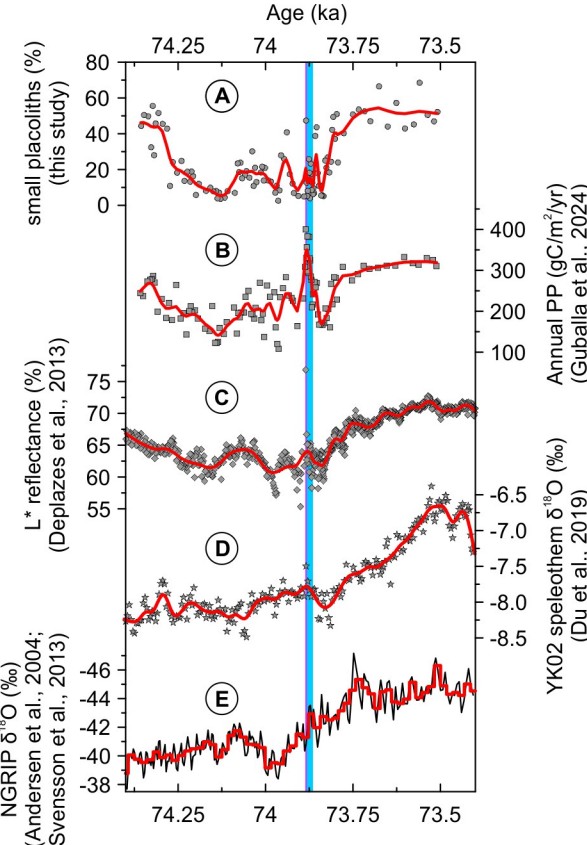

**Fig 5. Comparison of the coccolithophore assemblage to paleo-proxy records between 74.40 to 73.40 ka.** (A) small placoliths consisting of *E. huxleyi* and *G. ericsonii*. (B) Annual marine primary productivity (PP) record from core SO130-289KL [11]. (C) L* reflectance record [50]. (D) Stable oxygen isotopic composition ($\delta^{18}O$) of Yangkou YK07 speleothem from southwestern China [27]. (E) North Greenland Ice Core Project (NGRIP) $\delta^{18}O$. The red line is the 20-year mean [21] while the black line is a high-resolution record (4-year resolution) spanning the YTT eruption [88]. In panels A–D, the red lines show trends obtained from running a bootstrap (n = 1000) LOESS smoothing with a 0.1 ka window. The blue rectangle delineates the duration of the miniature slump fold preserved in core SO130-289KL. The violet line indicates the age of the YTT eruption (73.9 ka).

another (i.e., last glacial period) [92] might explain the occurrence of the "pseudo-stadial" assemblage. While the $\delta^{18}O$ signature of speleothems do not show similar variability (Fig 5D) [27], loess deposits from northern China show abrupt increases in median grain sizes indicating a strengthened East Asian winter monsoon just before the MIS 5a to MIS 4 transition [87, 93]. Therefore, coccolithophore assemblages appear to record not only millennial-scale D–O climate oscillations in the Arabian Sea but also accurately reflect abrupt decadal climate variability.

## Factors affecting upper photic zone coccolithophore assemblage

The variability of the overall coccolithophore assemblage in the northeastern Arabian Sea is associated with variations in the strength of the monsoons and related changes in the surface ocean mixed layer depth [72, 74]. Although mixed layer deepening has been shown to control the ratio of upper photic zone and lower photic zone species in marine sediments [11], there might be additional parameters that contribute to the observed relative abundance changes specifically of upper photic zone species.

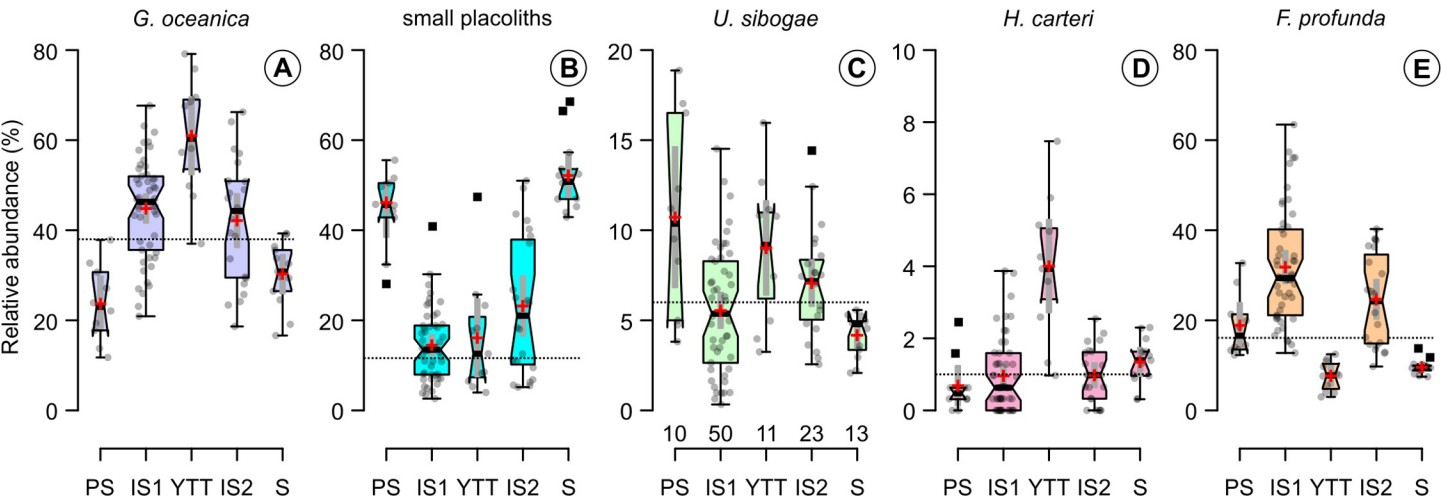

**Fig 6. Box-whisker plots of the relative abundances (%) of coccolithophore species characteristic for the "pseudo-stadial" (PS; 1854.6–1852.8 cm), the interstadial before the YTT eruption (IS1; 1852.6–1843.3 cm), immediately after the YTT eruption (YTT; 1843.1–1842.1 cm), the interstadial after the YTT eruption (IS2; 1842–1836.6 cm), and the stadial (S; 1835–1820.4 cm).** In all cases, n = 10 for PS, n = 50 for IS1, n = 11 for YTT, n = 23 for IS2, and n = 13 for S. (A) *G. oceanica*. (B) small placoliths consisting of *E. huxleyi* and *G. ericsonii*. (C) *U. sibogae*. (D) *H. carteri*. (E) *F. profunda*. Note different y-axis scales. The lower and upper bounds of the boxes represent the first and third quartiles of the data, respectively (i.e., the interquartile range or IQR), and the median is shown as a heavy horizontal line in the box. Notches approximate the 95% confidence interval of the median and are inverted when the 95% confidence interval exceeds the IQR. The red cross and thin grey bar show the mean and 95% confidence interval of the mean, respectively. Tukey style whiskers encompass all data within 1.5 times the IQR. Gray circles represent individual data points while black squares are outliers (i.e., values greater than 1.5 times the IQR). The dotted horizontal line in each panel shows the annual mean relative abundance of each species based on the sediment trap data of Andruleit et al. [72].

*Gephyrocapsa oceanica* is currently the most abundant coccolithophore species in the Arabian Sea as shown by plankton [74, 75, 94], sediment trap [72, 73], and surface sediment studies [95, 96]https://www.zotero.org/google-docs/?SMea51. Sediment trap data near our core location show maxima in both proportions and coccolith flux of *G. oceanica* during winter [72], concurrent with a deepened mixed layer and increased primary productivity (PP) (Fig 4D) [70, 71]. However, *G. oceanica* is composed of at least two distinct morphotypes—possibly different species—each with unique environmental preferences [97–100]. In particular, the *Gephyrocapsa* "Larger" morphotype was reported to occur in nutrient-rich upwelling sites and temperatures ranging from ~18–23°C [94, 97] while the *Gephyrocapsa* "Equatorial" morphotype seems to prefer warm water regions (~25–29°C) [99]. Thus, the assumption that increases in *G. oceanica* reflect solely nutrient-rich conditions [72, 74, 95, 96] might not be entirely correct. For example, we find a clear increase in *G. oceanica* concurrent with a deepened mixed layer right after the YTT eruption (Figs 2C and 6A). However, during the succeeding stadial characterized by a similar water column structure, small placolith species (i.e., *E. huxleyi* and *G. ericsonii*) are more abundant (Figs 2D and 6B).

Furthermore, the interpretation of the abundance changes in *H. carteri* in our record is also not straightforward as the ecology of *H. carteri* is not well established. In modern oceans, the distribution of *H. carteri* suggests a preference for warm waters [32, 33]https://www.zotero.org/google-docs/?Yp2nAy, turbid conditions [101, 102], and increased nutrient availability [95, 102, 103]. This species was also considered opportunistic due to its presence in estuarine environments with elevated nutrient availability [104, 105]. The preference of *H. carteri* to increased nutrient availability appears to be corroborated by our data right after the YTT eruption (Fig 2F and 2H), although *H. carteri* was low (<3%) during the succeeding stadial (Fig 6D).

In summary, the observed shifts within the composition of the upper photic zone coccolithophore assemblage may be related to additional factors rather than variations in the mixed

layer depth alone. Other processes such as dust input, ash fertilization, light reduction for photosynthesis, regional sea surface temperature changes, and evolutionary adaptation may have contributed to the observed changes in the upper photic zone coccolithophore assemblage in core SO130-289KL.

For instance, the short-lived ($\sim$2-year duration) and sharp increase of small placolith species comprised of *E. huxleyi* and *G. ericsonii* from 5 ± 2% just below the YTT layer to 47 ± 6% immediately above the YTT layer (Figs 2D and 6B) and their increased relative abundance in the succeeding cold stadial might be related to atmospheric deposition of ash and dust. Carbonate dissolution caused by the ash deposition [106] can be excluded as a main mechanism since this process should lead to an increase of the dissolution-resistant species *F. profunda* [107], contrary to what was previously observed in our record (Figs 2G and 6E) [11]. Thus, it is likely that the increase in small placoliths is a primary response to ash deposition concomitant with the YTT eruption, which might have stimulated the production of phytoplankton as observed for modern-day changes in phytoplankton associated with volcanic ash deposition [108–110]. This mechanism is comparable to modeling studies pointing towards dust deposition as an important factor in stimulating marine primary productivity in the present-day Arabian Sea [111, 112] that was also proposed for the Oman upwelling margin during the Last Glacial Maximum [56, 113]. Therefore, surface nutrient enrichment from dust deposition might have also commenced in earlier glacials and stadials that are, in general, characterized by increased dust fluxes [48, 114, 115]. Both ashfall and dust input might have resulted in the growth of small placolith species through micronutrient (e.g., iron) fertilization [108–110] although it is also important to note that culture experiments exploring the effects of volcanic ash on growth rates of *E. huxleyi* showed variable responses [116, 117].

Another mechanism that might have contributed to the increase in small placoliths is the change in light availability for photosynthesis. The YTT eruption produced aerosols that must have increased the stratospheric aerosol optical depth (AOD) of the atmosphere [7]. The increased AOD may have reduced light intensities available for photosynthesis and hence may have fostered growth of low light tolerant species. In addition, the ashfall may have increased surface water turbidity that may have further reduced light availability for a short period of time. Small placolith species such as *E. huxleyi* seems to adapt to a wide range of light levels [118] and thus might have gained advantage over other coccolithophore species from reduced light levels immediately after the YTT eruption. Likewise, the potential increase in dust input during stadials [48, 114, 115] might have also contributed to increased surface water turbidity and might explain the increase in small placolith species during the succeeding stadial.

The upper photic zone coccolithophore assemblage could have also been impacted by potential changes in sea surface temperatures (SST). The surface temperature effect of the YTT eruption is still debated, ranging from a cooling of 2–4˚C [7] to a potential warming of 2˚C [8] in models, and negligible (0.3 ± 1.5˚C; [31]) to significant cooling (1˚C; [28]) in alkenone SST proxies. However, the succeeding stadial was characterized by a cooling of $\sim$2.5˚C from 27˚C to 24.5˚C in the northeastern Arabian Sea [31] concomitant with the increase in small placolith species in our record (Figs 2D and 6B). Although small placolith species (i.e., *E. huxleyi*) are known to occur over a wide range of temperatures [33, 35, 119], culture experiments suggest that their growth rates are optimal at $\sim$17–20˚C [120, 121]. In contrast, *G. oceanica* is favored by warm SST conditions (> $\sim$22–25˚C) [99, 120] and is most abundant in marginal seas [96] along the West Pacific and in the Red Sea where the temperature ranges from 19–30˚C (mean temperature during plankton sampling = 27.2˚C) [122]. The relatively large drop in SST ($\sim$2.5˚C) during the succeeding stadial might have been sufficient to shift *G. oceanica* out of its optimum temperature conditions and be replaced by the small placolith species.

We also cannot simply exclude the effect of evolutionary adaptation on the changes in upper photic zone coccolithophores. In particular, the reported dominance (i.e., >50%) reversal from *G. caribbeanica* to *E. huxleyi* in mid-latitude regions (73 ka) [123] is close to the timing of the YTT eruption (73.9 ka). Indeed, shifts considered equivalent to this dominance reversal (i.e., from *G. ericsonii* to *E. huxleyi*) were also identified in the Arabian Sea [55], the tropical Indian Ocean [124, 125] and off the coast of South Africa [126, 127]. However, our data show that over the entire study interval the small placolith assemblage is mostly composed of *E. huxleyi* (S2 Fig) even though the species identification on the light microscope is limited. Our results suggest that (1) the reported dominance reversal might have already occurred and (2) the dominance reversal and the YTT eruption are independent events. Nevertheless, the evolutionary replacement of *G. caribbeanica*/*G. ericsonii* to *E. huxleyi* [123] may partly explain why *E. huxleyi* is more abundant than *G. ericsonii* in our record.

## Interdecadal cycles in the coccolithophore record during GI-20

Further insights on the potential factors affecting the coccolithophore assemblage in our record may be revealed by spectral analysis. Spectral analysis of the coccolithophore record before the YTT eruption indicates the presence of an interdecadal band ($\sim$20–22 and 29 years) using redfit, a method designed for unevenly spaced data (Fig 3). We also implemented additional frequency analysis on linearly interpolated data (S1 Text) that revealed broadly comparable results (S3 Fig) although these frequencies are only significant in a particular time frame (S4 Fig). There also seems to be a lower frequency band in our data (i.e., 60–70-year cyclicity) (S5 and S6 Figs) but this band was not consistently detected (Figs 3 and S3 and S4). Thus, we investigated whether other highly temporally resolved proxy records show frequencies at or near these interdecadal bands.

Indeed, the L* reflectance record [38] shows broadly comparable cyclicities at the 16–20-year band (S7A and S7D Fig). We also detected significant bands at 29–30 years, 19 years, and 15–16 years in the North Greenland Ice Core Project (NGRIP) $\delta^{18}$O record (S7B and S7E Fig) [88]. On the other hand, the power spectrum of the speleothem $\delta^{18}$O record [27] appears to be more complex and does not show a consistent presence of a statistically significant frequency band (S7C and S7F Fig). Moreover, these records do not show a consistent 60–70-year band (S7 Fig) but appear to exhibit a 20–30-year cyclicity similar to our data (Fig 3). The detection of broadly comparable cyclicities in ice [88], marine ([38] and this study) and, possibly, continental records [27] in the 20–30-year band (Figs 3 and S3 and S7) suggests that a global-scale mechanism might have driven the observed cycles although the exact process currently remains uncertain.

Ice core $\delta^{18}$O records from Greenland over the past millennium (1303–1961 CE) [128] and over the past 8 ka [129] revealed a 20-year cyclicity associated with variations in North Atlantic SST and the Atlantic Meridional Overturning Circulation (AMOC) that was also supported by climate models [130–132]. It is generally accepted that fluctuations in the strength of the AMOC occur synchronously (within age uncertainties; [133]) with changes in Greenland temperatures [21] and atmospheric circulation in the high latitude North Atlantic [18]. Therefore, we speculate that rapid atmospheric teleconnections might explain the presence of broadly comparable spectral bands found in continental [27], ice [88], and marine records ([38] and this study) (Figs 3 and S7).

However, other processes of similar frequencies can also be associated with the observed cyclicity in our record. For example, mechanisms proposed for bidecadal periodicities found in Holocene and Last Glacial Maximum proxy records include solar forcing [134, 135] and SST variations in the tropical–extratropical Pacific (Pacific Decadal Oscillation or PDO) [136,

137]. Another mechanism is the 18.6-year lunar nodal cycle (near the 20-year periods), as suggested for Late Pleistocene varve thickness data in the Arabian Sea [138] that apparently might also be related to large-scale climatic oscillations such as the El Niño–Southern Oscillation [139] and PDO [140]. On the other hand, the 29-year peak for *U. sibogae* might be related to a 31-year peak found in both gray values and varve thickness from a Holocene marine sediment core in the northeastern Arabian Sea linked to an unknown forcing that likely influenced both terrigenous flux and anoxia/productivity [134].

## Conclusions

We report an ∼850-year coccolithophore assemblage record from core SO130-289KL in the northeastern Arabian Sea (Fig 1), revealing previously unreported responses of upper photic zone coccolithophore species to the ∼73.9 ka Youngest Toba Tuff (YTT) supereruption and the transition from the warm Greenland Interstadial 20 (GI-20) to the cold Greenland Stadial 20 (GS-20). Our results indicate that:

1. Approximately 90% of the coccolithophore assemblage is composed of six species: *G. oceanica*, *U. sibogae*, *H. carteri*, *F. profunda*, and small placoliths belonging to *E. huxleyi* and *G. ericsonii* (Fig 2). Sediments right after the YTT eruption are characterized by increased proportions of *G. oceanica* and *H. carteri*, while small placoliths increased only in one sample immediately above the YTT layer. Sediments from GI-20 are mainly composed of *G. oceanica* (mean ∼41%) and *F. profunda* (mean ∼30%), while sediments from GS-20 are characterized (mean ∼52%) by the small placoliths (Figs 2 and 6).

2. Although the YTT eruption had a significant impact on the coccolithophore assemblage, our results support the view that the eruption was not the trigger for the GI-20 to GS-20 transition. A few decades after the YTT eruption, the coccolithophore assemblage shows a return to an "interstadial" composition whose major proportions are composed of *G. oceanica* and *F. profunda* before being replaced by small placoliths in the succeeding stadial (Figs 2 and 6).

3. Our results indicate the potential of coccolithophore assemblage changes as a tracer of Dansgaard–Oeschger (D–O) climate transitions in middle to low latitudes. During the interstadial (GI-20) to stadial (GS-20) transition, the change in the coccolithophore assemblage composition from *G. oceanica* and *F. profunda* to small placoliths composed of *E. huxleyi* and *G. ericsonii* aligns with many other proxies that recorded this climatic transition (Fig 5). The changes in the upper photic zone coccolithophore assemblage seem to be related to other factors rather than variations in mixed layer depth alone. Potential mechanisms include atmospheric ash and dust deposition and associated light reduction, cooling of sea surface temperature, and evolutionary adaptation.

4. Spectral analysis of data before the YTT eruption revealed an interdecadal variability (∼19–22 years, 29 years) (Fig 3) that compare well with other high temporally resolved proxies in the same time interval, as well as Holocene and Last Glacial Maximum proxy records. While the exact mechanism remains unknown, a global scale process—such as fast high- to low-latitude atmospheric teleconnections—might potentially transmit such signals to geographically widespread proxy records.

## Supporting information

**S1 Text. Additional frequency analyses using linearly interpolated data.**
(DOCX)

**S1 Fig. Scanning electron microscope (SEM) images of common coccolithophore species identified in the study interval in core SO130-289KL.** In all panels, the scale bar is 5 μm. (A) *Calcidiscus leptoporus*, 1842.7 cm core depth. (B) *Emiliania huxleyi*, 1842.7 cm core depth. (C) *Gephyrocapsa ericsonii*, 1843.1 cm core depth. (D) *G. oceanica*, 1843.7 cm core depth. (E) *Florisphaera profunda* var. *elongata*, 1843.1 cm core depth. (F) *F. profunda* var. *profunda*, 1840.7 cm core depth. (G) *Helicosphaera carteri*, 1843.1 cm core depth. (H) *Umbilicosphaera sibogae*, 1842.2 cm core depth.
(TIF)

**S2 Fig. Relative abundance (%) of the small placolith components in the studied interval of core SO130-289KL.** (A) Sediment core scan from Deplazes et al. [49] showing clearly visible YTT layers. (B) *E. huxleyi*. (C) *G. ericsonii*. Color bands for each species demarcate the 95% confidence interval for the relative abundances. The gray band shows the miniature slump fold enclosed by two ash layers from the YTT eruption, with the axial trace delineated by the red dotted line.
(TIF)

**S3 Fig. Multitaper spectral analysis of the main coccolithophore species in core SO130-289KL.** (A) *G. oceanica*. (B) small placoliths consisting of *E. huxleyi* and *G. ericsonii*. (C) *U. sibogae*. (D) *F. profunda*. The black line shows the spectrum for each species. Brown, orange, and red lines represent the 90%, 95%, and 99% confidence limits, respectively. The yellow rectangles highlight bands exceeding the 90% confidence limit. Yellow rectangles with asterisks show bands that are very close to the 90% confidence limit. The horizontal line in (B) shows the bandwidth (BW) resolution.
(TIF)

**S4 Fig. Wavelet analysis of the major coccolithophore species in core SO130-289KL.** A. *G. oceanica*. B. small placoliths consisting of *E. huxleyi* and *G. ericsonii*. C. *U. sibogae*. D. *F. profunda*. The heavy black contour lines denote the 95% significance level. Gray shaded regions show the cone of influence, where edge effects become important and thus should be interpreted with caution (i.e., the calculation of the spectral variance lies outside of the data boundaries).
(TIF)

**S5 Fig. Ensemble empirical mode decomposition (EEMD) analysis of the major coccolithophore species in core SO130-289KL.** In all panels, four intrinsic mode functions (IMF) show oscillations inherent to the relative abundance data, along with a residual. The EEMD analysis was done using both raw (black lines) and interpolated (red lines) relative abundance data. A. *G. oceanica*. B. small placoliths consisting of *E. huxleyi* and *G. ericsonii*. C. *U. sibogae*. D. *F. profunda*.
(TIF)

**S6 Fig. The addition of the residual with the second intrinsic mode function (IMF) based on ensemble empirical mode decomposition (EEMD) analysis to place the EEMD results in context with the relative abundance data.** In all panels, the relative abundance data (black lines) are plotted with the residual + IMF2 (red lines) for both interpolated (A)

and raw (B) datasets.
(TIF)

**S7 Fig. Spectral analyses of highly temporally resolved proxy records using the "redfit" (A-C) and multitaper (D-F) methods.** (A) and (D): L* reflectance of core SO130-289KL [38]. (B) and €: North Greenland Ice Core Project (NGRIP) $\delta^{18}$O record [88]. (C) and (F): The YK07 speleothem stable oxygen isotope ($\delta^{18}$O) record from southwest China [27]. The black line shows the spectrum for each proxy record. Brown, orange, and red lines represent the 90%, 95%, and 99% confidence limits, respectively. The yellow rectangles highlight bands exceeding the 90% confidence limit. Yellow rectangles with asterisks show bands that are very close to the 90% confidence limit. The horizontal line in (F) shows the bandwidth (BW) resolution for the multitaper analysis.
(TIF)

**S1 Table. Coccolithophore relative abundance data from core SO130-289KL from 1854.6 cm to 1820.4 cm.**
(XLSX)

## Acknowledgments

We gratefully acknowledge the scientific team and the shipboard captain and crew of the RV SONNE 130 for the acquisition of samples that made this study possible. We are grateful to the editor, Dr. Alessandro Incarbona, Dr. Mário Cachão and an anonymous reviewer whose comments significantly improved our paper.

## Author Contributions

**Conceptualization:** Jose Dominick Guballa, Jörg Bollmann, Katherine Schmidt.

**Data curation:** Jose Dominick Guballa.

**Formal analysis:** Jose Dominick Guballa, Katherine Schmidt.

**Funding acquisition:** Jörg Bollmann.

**Investigation:** Jose Dominick Guballa, Katherine Schmidt.

**Methodology:** Jörg Bollmann, Katherine Schmidt.

**Project administration:** Jose Dominick Guballa, Jörg Bollmann.

**Resources:** Jörg Bollmann, Andreas Lückge.

**Supervision:** Jörg Bollmann.

**Validation:** Jose Dominick Guballa, Jörg Bollmann.

**Visualization:** Jose Dominick Guballa.

**Writing – original draft:** Jose Dominick Guballa, Jörg Bollmann.

**Writing – review & editing:** Jose Dominick Guballa, Jörg Bollmann, Katherine Schmidt, Andreas Lückge.

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
