## [Decision Letter · Decision Letter 0]

18 Jul 2024

PONE-D-24-25299Coccolithophore assemblage composition during the Greenland Interstadial­–Stadial 20 transition and their response to the Youngest Toba Tuff (YTT) supereruption ~74,000 years ago in the northeastern Arabian SeaPLOS ONE

Dear Dr. Guballa,

Thank you for submitting your manuscript to PLOS ONE. After careful consideration, we feel that it has merit but does not fully meet PLOS ONE’s publication criteria as it currently stands. Therefore, we invite you to submit a revised version of the manuscript that addresses the points raised during the review process.

**Both reviewers much appreciated your work, and i share this feeling. However, there is something to be improved, before publication. Please pay special attention to comments by Reviewer#2, who disputes spectral analysis results. Moreover, in my opinion U. sibogae paleoenvironmental indication is overlooked. Following best literature (you fully cite), U. sibogae shows a positive response, both recorded in relative and absolute abundances, to the onset of the NE monsoon (Andruleit et al., 2000). The cluster analysis carried out on surface sediments associates this species to *G. oceanica*, within a group characterized by environmental conditions with high nutrient availability (Andruleit and Rogalla, 2002). Thus, this would match your reconstruction, especially at YTT.**

We look forward to receiving your revised manuscript.

Kind regards,

Alessandro Incarbona

Academic Editor

PLOS ONE

Journal Requirements:

4. We noted in your submission details that a portion of your manuscript may have been presented or published elsewhere. Yes, data of the lower photic zone species Florisphaera profunda and the species Helicosphaera carteri was taken from a recently published work by Guballa et al. (2024) (Guballa JD, Bollmann J, Schmidt K, Lückge A. The Toba Eruption 74,000 Years ago Strengthened the Indian Winter Monsoon-Evidence From Coccolithophores. Paleoceanography and Paleoclimatology. 2024;39: e2023PA004823. doi:10.1029/2023PA004823).

Our new work is not a dual publication since we explore the response of the upper photic zone coccolithophore assemblages to the Toba eruption and an interstadial-stadial transition. Guballa et. al (2024) solely focused on the reconstruction of primary productivity and the impact of the Toba eruption using the relative abundance of the lower photic zone species Florisphaera profunda (i.e., upper versus lower photic zone species ratio only) for a PP transfer function. The species Helicosphaera carteri was used only to make assumptions about sedimentation processes in the core.

Furthermore, in our current manuscript we employ time series analysis (i.e., frequency analysis) to our data set to determine the mechanisms driving the variation of coccolithophore assemblage composition in the upper photic zone in the Arabian Sea.

We believe that all these make the current manuscript a new and unique contribution. Please clarify whether this [conference proceeding or publication] was peer-reviewed and formally published. If this work was previously peer-reviewed and published, in the cover letter please provide the reason that this work does not constitute dual publication and should be included in the current manuscript.

5. We note that Figure 1 in your submission contain map images which may be copyrighted. All PLOS content is published under the Creative Commons Attribution License (CC BY 4.0), which means that the manuscript, images, and Supporting Information files will be freely available online, and any third party is permitted to access, download, copy, distribute, and use these materials in any way, even commercially, with proper attribution. For these reasons, we cannot publish previously copyrighted maps or satellite images created using proprietary data, such as Google software (Google Maps, Street View, and Earth). For more information, see our copyright guidelines: http://journals.plos.org/plosone/s/licenses-and-copyright.

We require you to either present written permission from the copyright holder to publish these figures specifically under the CC BY 4.0 license, or remove the figures from your submission:

Reviewers' comments:

Reviewer's Responses to Questions

**Comments to the Author**

1. Is the manuscript technically sound, and do the data support the conclusions?

Reviewer #1: Yes

Reviewer #2: Partly

2. Has the statistical analysis been performed appropriately and rigorously? 

Reviewer #1: Yes

Reviewer #2: N/A

3. Have the authors made all data underlying the findings in their manuscript fully available?

Reviewer #1: Yes

Reviewer #2: Yes

4. Is the manuscript presented in an intelligible fashion and written in standard English?

Reviewer #1: Yes

Reviewer #2: Yes

5. Review Comments to the Author

**Reviewer #1:** I have no major criticisms of your manuscript, which I generally find to be very good. I have made a few comments on the attached copy.

You have correctly identified the impact of YTT as independent of the more regional climatic changes (interstadial-stadial) that occurred in the region.

I should have emphasized more the potential dust fertilization that you briefly mention in the discussion, giving it the prominence it deserves. For instance, you could show a more detailed graphic of the major coccolithophore components shortly before (e.g., depths: 1845.6 to 1843.3), during (depths: 1843.1 to 1842.1), and shortly after (e.g., depths: 1842.0 to 1837.6) the event.

You could also have included (for the record) results for some of the less abundant species, like C. leptoporus, which interestingly did not show much change. For the other less abundant species, of course additional counts should have been made, particularly if your count of 300 includes small placoliths and/or F. profunda.

**Reviewer #2:** The manuscript "Coccolithophore assemblage composition during the Greenland Interstadial–Stadial 20 transition and their response to the Youngest Toba Tuff (YTT) supereruption ~74,000 years ago in the northeastern Arabian Sea" by Guballa et al. presents a high-resolution dataset aimed at enhancing the understanding of coccolithophores during the stadial/interstadial transition and the YTT supereruption. Statistical and signal analyses are also used to elucidate and improve the primary relationships between coccolithophores and environmental factors.

The authors describe the data in great detail and illustrate their findings with an adequate number of high-quality figures. The references are up-to-date.

I fully agree with the authors' conclusion that “the eruption was not the trigger for the GI-20 to GS-20 transition.”

All my comments and suggestions are intended to improve the quality of the paper, which needs clarification in some parts of the text.

Line 22: “annually resolved” – The mean resolution is 7.85 years, and for the first 60 samples (before YTT), it is 7.9 years. I strongly suggest using “decadal resolved” not only in line 22 but throughout the text, particularly in the “Materials and Methods” section.

“Spectral analyses”: This is a critical point. The authors, using “redfit,” find a ~20-year periodicity in the first 60 samples (before YTT), but do not check (e.g., wavelet analyses) if this periodicity is consistently present throughout the interval. I have performed some checks: interpolating samples at 7.9 years (no artificial decreases were observed) and performing wavelet analysis, the ~20-year periodicity is barely visible only in the first 10 samples of the interval. Consequently, I think that this 20-year periodicity is negligible and useless (probably a multiple of the 11-year periodicity but impossible to check due to the “low” resolution). Conversely, applying the Ensemble Empirical Mode Decomposition algorithm, I have found very interesting results in the IMF. Notably, there is a ~70-year periodicity in G. oceanica and E. huxleyi, showing a continuous anti-correlation (in line with their respective ecological preferences) in cross-wavelet analyses. I strongly suggest performing new signal analyses, especially since the authors noticed the effect of the 70-year periodicity (line 330) but did not fully explore the potential of this signal.

Line 126: “using coccolithophore species that were consistently present” – Please specify the taxa.

Line 384: “the interpretation of the abundance changes in H. carteri …..” – In my opinion, the authors report interesting data on H. carteri. I suggest referencing Bonomo et al., Marine Micropaleontology, https://doi.org/10.1016/j.marmicro.2021.101995 (see station 25). This paper reports (living coccoliths) an interesting and comparable result for H. carteri. Notably, the abundance of H. carteri (inside the mini slump interval) vs. small placoliths (E. huxleyi) shows comparable results, indicating that the coccolithophore assemblages exhibited an inversion of dominance, characterized by low E. huxleyi vs. relatively high abundance of H. carteri.

Line 453 “Interdecadal cycles in the coccolithophore record during GI-20”: This chapter must be completely revised based on new signal analyses (see above) and consequently, the “Conclusion” chapter.

6. PLOS authors have the option to publish the peer review history of their article (what does this mean?). If published, this will include your full peer review and any attached files.

Reviewer #1: **Yes: **Mário Cachão

Reviewer #2: No

---

## [Author Response · Author response to Decision Letter 0]

20 Aug 2024

August 19, 2024

Dear Dr. Incarbona, 

We appreciate the helpful comments and suggestions from you and the reviewers that significantly improved our manuscript. We have completed our responses to the reviewers and highlight in this letter the major points we addressed.

We emphasize that we report relative abundances that are bounded by the "unit-sum constraint", where the loss of one species automatically results in the gain of another species as the sum of all species present must be constant. For instance, the increase in Umbilicosphaera sibogae right after the Toba eruption—which you and Reviewer #1 pointed out—might be driven by the loss of other species. In addition, U. sibogae in the northeastern and northwestern Arabian Sea show different responses to increased nutrient availability, highlighting the ecological complexity of this species. Thus, we opted to not focus on the ecological indications of U. sibogae in our data.

Based on Reviewer #2’s comments, we now provide additional frequency analyses that we believe strengthened our initial argument of the presence of a 20–30-year band. These analyses indicate a more consistent presence of the 20–30-year band both in our data and in other proxy records in contrast to the ~70-year cyclicity as observed by Reviewer #2. Although the 70-year cyclicity as suggested by Reviewer #2 is a very interesting observation, it is not corroborated by any other proxy record nor statistical analysis that such a cyclicity exists in our record.

As we have noted in our initial submission, some of the data (i.e., relative abundance of Florisphaera profunda and Helicosphaera carteri) was taken from a recently published work (Guballa JD, Bollmann J, Schmidt K, Lückge A. The Toba Eruption 74,000 Years ago Strengthened the Indian Winter Monsoon-Evidence From Coccolithophores. Paleoceanography and Paleoclimatology. 2024;39. doi:10.1029/2023PA004823). Our current manuscript is not a dual publication of the same content since we explore the response of the upper photic zone coccolithophore assemblages to an interstadial-stadial transition and to the Toba eruption, which was not explored previously. Guballa et al. (2024) solely focused on the reconstruction of primary productivity and the impact of the Toba eruption using the relative abundance of F. profunda (i.e., upper versus lower photic zone species ratio only) to reconstruct primary productivity. The species H. carteri was used only to make assumptions about sedimentation processes in the core. Furthermore, in our new work we employ time series analysis (i.e., frequency analysis) to determine the mechanisms driving the variation of the coccolithophore assemblage composition in the upper photic zone in the Arabian Sea.

We believe that all these make the current manuscript a new and unique contribution.

Thank you very much for considering our work for publication in your journal. We hope our manuscript now meets the standards of PLOS ONE for publication. 

Sincerely, on behalf of all the authors,

Jose Dominick Guballa

---

## [Editor Report · Decision Letter 1]

23 Aug 2024

Coccolithophore assemblage composition during the Greenland Interstadial­–Stadial 20 transition and their response to the Youngest Toba Tuff (YTT) supereruption ~74,000 years ago in the northeastern Arabian Sea

PONE-D-24-25299R1

Dear Dr. Guballa,

We’re pleased to inform you that your manuscript has been judged scientifically suitable for publication and will be formally accepted for publication once it meets all outstanding technical requirements.

Kind regards,

Alessandro Incarbona

Academic Editor

PLOS ONE

---

## [Editor Report · Acceptance letter]

28 Aug 2024

PONE-D-24-25299R1 

PLOS ONE

Dear Dr. Guballa, 

I'm pleased to inform you that your manuscript has been deemed suitable for publication in PLOS ONE. Congratulations! Your manuscript is now being handed over to our production team.

Kind regards, 

on behalf of

Professor Alessandro Incarbona 

Academic Editor

PLOS ONE